# Healthcare providers' awareness of a mobile-based app for disease classification in northwest Ethiopia: A multilevel logistic regression analysis

Adamu Takele Jemere[1]*, Tesfahun Melese Yilma[1], Lemma Derseh Gezie[2], Shegaw Anagaw Mengiste[3], Monika Knudsen Gullslett[4], Jens Johan Kaasbøll[5], Binyam Tilahun[1]

**1** Department of Health Informatics, Institute of Public Health, College of Medicine and Health Sciences, University of Gondar, Gondar, Ethiopia, **2** Department of Epidemiology and Biostatistics, Institute of Public Health, College of Medicine and Health Sciences, University of Gondar, Gondar, Ethiopia, **3** Management Information Systems, University of South-Eastern Norway, Drammen, Norway, **4** Norwegian Centre for eHealth research, University hospital of Northern Norway, Tromsø, Norway, **5** Health Information System Programme Centre, University of Oslo, Oslo, Norway

\* adamutakele@gmail.com

## Abstract

### Background

Mobile health (mHealth) apps improve healthcare providers' accurate disease classification in resource-limited settings. Ethiopia recently introduced the National Health Data Dictionary (NHDD) mobile app for disease classification; however, healthcare providers' awareness of it remains unknown. This study aimed to assess awareness of mobile-based disease classification apps among healthcare providers working in public health facilities in northwest Ethiopia and to determine the factors associated with this awareness.

### Methods

A facility-based cross-sectional study was conducted among 423 healthcare providers working at 19 public health facilities in northwest Ethiopia from October 1 to 25, 2023. Data were collected using a pre-tested self-administered questionnaire. Awareness was defined as being aware of the existence of a mobile app (NHDD) for disease classification. Multilevel logistic regression analysis was used to account for clustering at the health facility level. Adjusted Odds Ratio (AOR) with 95% confidence interval (CI) was used to identify associated factors.

### Results

Only 30.73% (95% CI: 26.30%−35.55%) of healthcare providers were aware of a mobile-based app for disease classification. Healthcare providers having social media accounts (AOR = 13.96; 95% CI: 2.33–83.64), ever visited the medical field by

**Data availability statement:** All relevant data are within the manuscript and its Supporting information files.

**Funding:** The author(s) received no specific funding for this work.

**Competing interests:** The authors have declared that no competing interests exist.

a mobile phone (AOR = 2.39; 95% CI: 1.03–5.51), digital literacy (AOR = 6.13; 95% CI: 1.50–25.01), awareness of the ESV-ICD-11 booklet on paper (AOR = 2.34; 95% CI: 1.06–5.18), and access to ESV-ICD-11 training or mentorship (AOR = 2.93; 95% CI: 1.25–6.87) were factors associated with awareness.

## Conclusions

About one-third of healthcare providers are aware of the mobile-based disease classification app. Social media use, digital literacy, prior mobile use for the medical field, familiarity with the paper-based ESV-ICD-11 booklet, and ESV-ICD-11 training or mentorship were associated factors with awareness. Targeted awareness creation interventions could be considered to support the success of mobile-based app implementation in Ethiopia.

## Introduction

In recent years, the integration of mobile health (mHealth) technology has transformed healthcare, mainly in countries with limited resources, by enhancing disease diagnosis and aiding physicians in accurate decisions [1]. Among these technologies, mobile healthcare apps potentially improve clinical documentation, standardize data, provide real-time decision support, and enhance disease classification accuracy, ultimately contributing to more accurate and efficient healthcare services [2 - 4]. Particularly, disease classification applications enable medical professionals by standardizing classification and coding procedures and enhancing the completeness and quality of clinical data [5 - 7].

Despite the mobile apps' advantages, disparity in the use and awareness of these technologies persists. Research conducted in different regions indicates that greater use of mobile apps is correlated with healthcare providers' awareness of apps [8, 9]. However, lack of knowledge and digital literacy are reported as barriers, which have been observed by 39.6% and 38% of healthcare personnel, respectively, in studies [10, 11]. Research from countries like Malaysia and Australia has shown that healthcare professionals who are aware of mHealth are more likely to incorporate these tools into their regular clinical practice, indicating that awareness and familiarity with these tools have a significant impact on implementation outcomes [12 - 16].

Awareness of mobile-based apps is influenced several individual and facility-level factors. These include digital literacy, smartphone ownership, prior ICT training, work experience, age, education level, and rural residency [9, 17-20]. However, even among healthcare providers who are expected to use digital tools, there are ongoing gaps in awareness in many low-income nations, including Ethiopia [21]. Furthermore, raising awareness of mHealth can promote its integration into healthcare systems [22].

Ethiopia has created the National Health Data Dictionary (NHDD), a mobile application based on the Ethiopian Short Version of ICD-11 (ESV-ICD-11), to meet the demand for consistent disease classification [23]. This tool was created to assist

healthcare providers in classifying diseases in accordance with World Health Organization (WHO) recommendations [24]. Although there is a lot of potential for the app to improve the consistency and quality of clinical data, little is known about how well-informed healthcare providers are about its existence and capabilities.

By assessing healthcare providers' awareness of the NHDD mobile app in the Central Gondar Zone of northwest Ethiopia, this study aims to close that awareness gap. The study will help guide focused initiatives to raise awareness and eventually promote the broad use of mHealth tools for better disease classification by determining the present level of awareness as well as the related individual and facility-level characteristics.

## Materials and methods

### Study design, setting and period

A facility-based cross-sectional study was carried out in healthcare facilities situated across four districts in Ethiopia's Central Gondar zone from October 1 to 25, 2023. The Central Gondar Zone is 726 kilometers northwest of Addis Ababa, the capital of Ethiopia. According to the Zone Finance and Economic Department, there are 15 districts, serviced by 404 health posts, 10 public hospitals, and 75 health centers [25]. For this study, four districts, East Belesa, East Dembia, Tachi Armachiho, and West Dembia, were chosen randomly. All available healthcare facilities in the four districts (16 health centers and three primary hospitals) were included.

### Eligibility criteria

All healthcare providers who have been employed in Central Gondar Zone public health facilities for more than six months were included in this study. The study excluded health care providers who were seriously ill during the data collection period.

### Sample size determination and sampling procedure

The sample size was determined using the single population proportion formula ($n = z_{\alpha/2}p(1-P)/d^2$), considering a 95% confidence interval, 5% margin of error, 50% proportion of awareness of mobile-based app implementation for disease classification since no previous study, and 10% for non-response rate. The final calculated sample size was 423. From the total of 15 districts in the central Gondar zone, four districts were identified using simple random sampling. In the selected districts, available public healthcare facilities (16 health centers and 3 hospitals) were included. The total calculated sample size of 423 healthcare providers involved in disease coding and classification was allocated across the facilities proportionally.

### Study variables and operational definitions

**Dependent variable.** The dependent variable for this study was awareness of mobile-based apps for disease classification (NHDD).

**Independent variable.**

• Individual-level variables: Gender, age, profession, religion, educational status, marital status, type of profession, experience, salary, mobile phone access, type of phone, social media account ownership, type of social media, ICT training, ever visited the medical field on your mobile device, Digital literacy, and prior awareness of the ESV-ICD-11 booklet.

• Facility-level variables: Facility location, type of facility, and mentorship or training access.

**Operational definition.** Awareness of mobile-based apps for disease classification can be categorized as "aware" and "not aware" based on the question assessing healthcare providers' familiarity with the existence of the Ethiopian mobile-based app for disease classification (NHDD app).

## Data collection tools and procedures

Data were collected using a pre-tested, self-administered, and structured questionnaire. To collect data, the questionnaire was first created in English and then translated into Amharic. Later, for analysis, it was translated back into English from Amharic. A total of 10 bachelor's degree data collectors and 2 master's degree supervisors in health informatics were recruited to collect the data.

The dataset used in this study is provided as Supporting information (S1 Data).

## Data quality assurance

The quality of data was controlled starting from the questionnaire design stage. First, the questionnaire, which was prepared in English, was translated into the local language, Amharic, and then translated back to English for analysis to ensure the consistency of the tool. Before the data collection date, the data collectors and supervisors had two days of training on the data collection tools (contents of the questionnaire, interviewing techniques, purpose of the study, research ethics, approach to the interviewee, and confidentiality). A pretest was conducted on 5% of the sample, outside the study area, where the population shares similar characteristics, and modifications were made. The supervisors reviewed the questionnaires' completeness each day. Face and content validity were conducted using experts.

## Data processing and analysis

Following the completion of the data cleaning procedure and a thorough review of each completed questionnaire for missing values, the cleaned data was entered into Epi Info version 7 and exported to STATA Version 14 for statistical analysis. The data was described using descriptive statistics, which included frequency and cross-tabulation. To evaluate the variables associated with the app's awareness, a multilevel logistic regression was used because of the hierarchical data structure (healthcare providers nested within facilities). Candidate variables for multivariable modeling were found with a p-value of 0.2 or less using bi-variable analysis. Finally, significant factors were determined using the presented an adjusted odds ratio with its 95% CI and a P-value $<0.05$ was considered statistically significant.

Model-building was conducted in four steps using a random intercept approach: Model comparison was conducted to identify the best-fitted model out of four models and to identify and explain the factors.

Model 1 (Null Model): Without the use of individual or facility-level independent variables, this model assesses the degree of awareness variation between facilities. To justify the use of multilevel modeling, the intraclass correlation coefficient (ICC) was calculated.

Model II (The Individual-Level Model): Only individual-level variables (Level 1) were included in this model, including sex, age, educational status, occupational status, years of experience, monthly salary, ownership of a social media account, ICT training, mobile use for medical purposes, digital literacy, and awareness of the ESV-ICD-11 manual booklet.

Model III (Facility-Level Model): Only facility-level factors (Level 2) were included in this model, such as the facility location (urban/rural), type of healthcare facility (health center or hospital), and whether the healthcare facility offered mentorship or ESV-ICD-11 training or not.

Model IV (Combined Model): There were predictors at the facility and individual levels in this model. When clustering effects were taken into account, it was utilized to find key variables that were independently associated with awareness.

Model comparison was conducted to identify the best-fitted model for the factors.

## Ethics approval and consent to participate

The Declaration of Helsinki's ethical guidelines was followed in conducting this study. Ethical clearance was obtained from the Institutional Review Board of the University of Gondar College of Medicine and Health Science and Comprehensive Specialized Hospital (*Ref. no. ህጤና እጠ/ስፕ/ሆ/ምማኡ/06/02/06/9/2023*). The Amhara Public

Health Institute and the Zonal Health Department also provided a letter of support. Informed written consent was obtained from the participants after explaining the purpose of the study. To ensure confidentiality, their names and other personal identifiers was not registered. The study participants' participation in it was entirely voluntary. Participants had the chance to ask anything about the study as well as freedom to refuse or stop the interview at any time.

## Results

### Socio-demographic characteristics

In this study, 384 healthcare providers from Central Gondar Zone healthcare facilities participated, with a 90.78% response rate. These providers were spread throughout four districts: Tachi Armachiho, 117 (30.47%); East Dembia, 112 (29.17%); East Belesa, 87 (22.66%); and West Dembia, 68 (17.71%). In terms of age, 45.83% of participants were under 30, and the majority of participants were male (64.58%). Over half held a first degree (54.69%) and were married (57.55%) in terms of educational and marital status, respectively (Table 1).

**Table 1. Socio-demographic characteristics of healthcare providers' awareness of mobile-based app for disease classification in Central Gondar Zone healthcare facilities, 2023 (N = 384).**

| Variable | Category | Frequency | Percent |
|---|---|---|---|
| Facility type | Health Center | 220 | 57.29 |
|  | Primary Hospital | 164 | 42.71 |
| Sex | Female | 136 | 35.42 |
|  | Male | 248 | 64.58 |
| Age (in years) | <30 | 176 | 45.83 |
|  | 30-39 | 187 | 48.70 |
|  | >=40 | 21 | 5.47 |
| Religion | Orthodox | 370 | 96.35 |
|  | Muslim | 14 | 3.65 |
| Facility location | Urban | 274 | 71.35 |
|  | Rural | 110 | 28.65 |
| Educational Status | Diploma | 167 | 43.49 |
|  | Degree | 210 | 54.69 |
|  | Master | 7 | 1.82 |
| Marital Status | Single | 149 | 38.80 |
|  | Married | 221 | 57.55 |
|  | Divorced | 14 | 3.64 |
| Type of Profession | Nurse | 319 | 83.07 |
|  | Health Officer | 26 | 6.77 |
|  | Health Information Technician | 18 | 4.69 |
|  | Medical Doctor | 21 | 5.47 |
| Experience (in years) | <=5 | 152 | 39.58 |
|  | 6-10 | 177 | 46.09 |
|  | >10 | 55 | 14.32 |
| Salary (Ethiopian Birr) | <6000 | 98 | 25.52 |
|  | 6000-8000 | 165 | 42.97 |
|  | >8000 | 121 | 31.51 |

## Healthcare providers' awareness of mobile-based app for disease classification

Table 2 shows an overview of healthcare providers' awareness and use of the NHDD mobile app and the Ethiopian Short-Version ICD-11 (ESV-ICD11) booklet guide for disease classification in Central Gondar Zone public health facilities. While 60.42% of the providers were not familiar with the booklet guide, over half (55.73%) were aware of the ESV-ICD11's objective. Of those who knew, most (76.2%) said they used the booklet guide; however, 62.07% said they had difficulties. Just 30.73% of respondents aware of the NHDD mobile app, and 58.47% of them had used it.

## Technology-related factors

The majority of healthcare providers (72.51%) used smartphones, and nearly all (99.48%) had access to a mobile phone. A significant percentage (84.64%) had accounts on social media, mostly on Facebook (74.48%) and Telegram (71.35%). Even though the majority of providers (70.05%) lacked ICT training, most, 334 (86.98%) of healthcare providers reported as digitally literate (Table 3).

## Multilevel logistic regression analysis

To assess the factors related to healthcare providers' awareness of mobile-based app for disease classification in health care facilities, a multilevel logistic regression analysis was carried out. The health facility was designated as the level 2 variable in order to account for the clustering of healthcare providers within facilities.

**Model Fit statistics and comparison.** With an intra-class correlation coefficient (ICC) of 0.3439, the empty (null) model showed a significant clustering effect, indicating that facility differences accounted for about 34.4% of the overall variance in awareness. The Akaike Information Criterion (AIC) decreased from 405.78 in the null model to 339.29 in the full model (Model 4), indicating that the addition of individual- and facility-level covariates gradually enhanced model fit. The inclusion of both individual- and facility-level factors was supported by the statistical significance of the likelihood ratio test between Models 2 and 4 ($p = 0.0297$) (Table 4).

**Table 2. Healthcare providers' awareness of mobile-based app implementation for disease classification in Central Gondar Zone public health facilities, 2023 (N = 384).**

| Variable | Category | Frequency | Percent |
|---|---|---|---|
| Know the purpose of using ESV-ICD 11 | No | 170 | 44.27 |
| | Yes | 214 | 55.73 |
| Awareness of the ESV-ICD 11 booklet guide | No | 232 | 60.42 |
| | Yes | 152 | 39.58 |
| Use the ESV-ICD 11 booklet guide from those who are aware it | No | 37 | 24.34 |
| | Yes | 115 | 75.66 |
| Had a challenge when using the ESV-ICD 11 booklet guide from those who used it | No | 44 | 38.26 |
| | Yes | 71 | 61.74 |
| Awareness of the NHDD mobile app existence | No | 266 | 69.27 |
| | Yes | 118 | 30.73 |
| Use of the NHDD mobile app by those who are aware of it | No | 49 | 41.53 |
| | Yes | 69 | 58.47 |
| Had a challenge when using the NHDD mobile app from those who used it | No | 19 | 27.54 |
| | Yes | 50 | 72.46 |
| Receive mentorship or training on ESV-ICD 11 | No | 301 | 78.38 |
| | Yes | 83 | 21.62 |

**Note:** ESV-ICD11- Ethiopian Short Version International classification of disease version 11, NHDD= = National Health Data Dictionary

**Table 3. Technology related factors of healthcare providers' awareness of mobile-based app implementation for disease classification in Central Gondar Zone healthcare facilities, 2023 (N = 384).**

| Variable | Category | Frequency | Percent |
|---|---|---|---|
| Mobile Phone Access | Yes | 382 | 99.48 |
| | No | 2 | 0.52 |
| Type of Phone | Basic phone | 105 | 27.49 |
| | Smartphone | 277 | 72.51 |
| Social media account ownership | Yes | 325 | 84.64 |
| | No | 59 | 15.36 |
| Type of social media* | Imo | 216 | 56.25 |
| | Telegram | 274 | 71.35 |
| | Facebook | 286 | 74.48 |
| | WhatsApp | 28 | 7.29 |
| | LinkedIn | 3 | 0.78 |
| Take ICT training | Yes | 115 | 29.95 |
| | No | 269 | 70.05 |
| Have you ever visited the medical field on your mobile device | Yes | 248 | 64.58 |
| | No | 135 | 35.42 |
| Mobile phone use* | Calling or messaging | 337 | 87.76 |
| | Entertainment | 88 | 22.92 |
| | Clinical guideline | 212 | 55.21 |
| | Health-related information | 182 | 47.40 |
| Digital literacy | Literate | 334 | 86.98 |
| | Not literate | 50 | 13.02 |

*= The percent exceeds 100% since a participant may have more than one choice.

**Factors associated with awareness.** The multilevel logistic regression analysis identified individual and facility-level significant factors of healthcare providers' awareness of the mobile-based app for disease classification. Social media account ownership, ever visited the medical field by mobile phone, digital literacy, aware of the ESV-ICD-11 booklet on paper were individual-level factors while training or mentorship on ESV-ICD-11 was identified as facility-level factors.

Health care providers having social media accounts had substantially higher odds of awareness (AOR = 13.96, 95% CI: 2.33–83.64, $p < 0.01$) than those without social media accounts. Similarly, there was a higher likelihood of awareness among providers who had ever visited the medical field by mobile phone (AOR = 2.39, 95% CI: 1.03–5.51, $p < 0.03$). Additionally, digital literacy was a significant factor; healthcare providers who were digitally literate had six times the odds of being aware (AOR = 6.13, 95% CI: 1.50–25.01, $p < 0.05$) than those who were not. Health care providers who were aware of the ESV-ICD-11 booklet on paper were more likely to be aware of a mobile-based disease classification app (AOR = 2.34, 95% CI: 1.06–5.18, $p < 0.03$).

Similarly, nearly three times greater odds of awareness were related to healthcare providers' accessed training or mentorship on ESV-ICD-11 (AOR = 2.93, 95% CI: 1.25–6.87, $p < 0.05$) (Table 5).

## Discussion

According to this study, around one-third of central Gondar Zone healthcare providers are aware of the NHDD app, a mobile-based app for disease classification. The multilevel logistic regression analysis showed significant clustering of awareness across healthcare facilities (ICC = 34.4%), indicating the contextual differences. App awareness was

**Table 4. Model Fit Statistics and Comparison of multilevel models for healthcare providers' awareness of mobile-based app implementation for disease classification in Central Gondar Zone Healthcare Facilities, 2023.**

| Model | ICC | AIC | BIC | Likelihood Ratio Test (p-value) |
|---|---|---|---|---|
| Null Model (Model 1) | 0.3439 | 405.78 | 413.68 | — |
| Model 2 (Individual-level) | 0.3393 | 342.26 | 421.27 | — |
| Model 3 (Facility-level) | 0.2719 | 378.76 | 398.51 | — |
| Model 4 (Full Model) | 0.2835 | 339.29 | 430.15 | 0.0297 |

significantly associated with individual-level factors, including social media account ownership, prior use of mobile phones for medical purposes, digital literacy, and familiarity with the Ethiopian Short-Version ICD-11 booklet and a facility-level factor as receipt of ESV-ICD 11 related training or mentorship.

These results are consistent with various studies that found mHealth adoption was significantly associated with digital access and prior exposure to digital health factors [10, 11, 18]. The low level of awareness found in this study indicated areas where targeted action is required and highlights a substantial gap in Ethiopian healthcare providers' awareness of and use of digital health technologies. According to previous studies, implementing mHealth applications has been hindered by a lack of awareness and inadequate digital infrastructure [10, 22]. For example, one study indicated that healthcare professionals said only some were sufficiently aware of mHealth solutions, which results in a low use of apps for clinical purposes [10].

Conversely, awareness in this study is lower than the studies conducted in more digitally advanced healthcare settings such as England [26], Malaysia [15], India [27], Ghana [28], and other Ethiopian studies in urban areas [29, 30]. The reason may be attributed to the maturity of the digital system among developed settings, as well as the relatively recent introduction of our mHealth (NHDD) app, which is based on the ESV-ICD-11 disease classification standard. These findings highlight the importance of awareness-building as the first step toward successfully implementing innovations in digital health.

On the other hand, a study in Malaysia indicated low mHealth awareness (20.4%) [31] compared to our study. This difference might be due to differences in study populations, as our study conducted among healthcare providers, whereas the study in Malaysia conducted among the general population.

Social media account ownership was one of the factors that influenced healthcare providers' awareness of the NHDD app. Healthcare providers who owned social media accounts were almost 14 times more likely to be aware of the app. A systematic review and meta-analysis study showed social media greatly raises awareness of mHealth adding evidence to this finding [32]. This suggests that social media platforms are vital for disseminating information to enhance mHealth awareness like NHDD.

Similarly, prior experience on visiting medical field via a mobile phone played significant role in mobile-based disease classification app awareness levels. The odds of those who had ever visited the medical field via a mobile phone were 2.39 times more aware of the mobile-based disease classification app than those who did not. This result is in line with a previous study that showed general mHealth use experience was two times more likely to be aware of the specific intended mobile-based app [33]. This may be due to their familiarity with mHealth and perceived usefulness, as evidenced by a study in Ethiopia [34].

Digital literacy was also significantly associated with mobile-based disease classification app awareness. Healthcare providers with digital literacy skills were about six times more likely to be aware of the mobile-based disease classification app than those without such skills. This finding aligns with a study conducted in Germany, that indicated health professionals' eHealth literacy as the main barrier to implementation [35]. Another study also supported that digital literacy as one of the factors to ready to use mHealth solutions [36].

**Table 5. Multilevel Logistic regression analysis of healthcare providers' awareness of mobile-based app implementation for disease classification in Central Gondar Zone healthcare facilities, 2023 (N = 384).**

| Variable | Category | Not Aware, n (%) | Aware, n (%) | COR (95% CI) | AOR (95% CI) |
|---|---|---|---|---|---|
| Gender | Female | 106 (27.60) | 30 (7.81) | 1 | 1 |
| | Male | 160 (41.67) | 88 (22.92) | 2.53 (1.42-4.53) | 1.77 (0.85-3.69) |
| Age | <30 | 114 (29.69) | 62 (16.15) | 1 | 1 |
| | 30-39 | 136 (35.42) | 51 (13.28) | 0.77 (0.45-1.33) | 0.72 (0.33-1.57) |
| | >=40 | 16 (4.17) | 5 (1.30) | 0.88 (0.27-2.88) | 2.74 (0.35-21.53) |
| Educational Status | Diploma | 120 (31.25) | 47 (12.24) | 1 | 1 |
| | Degree | 144 (37.50) | 66 (17.19) | 1.11 (0.63-1.97) | 0.62 (0.29-1.34) |
| | Master | 2 (0.52) | 5 (1.30) | 8.32 (1.26-54.77) | 1.30 (0.15-10.89) |
| Marital Status | Single | 97 (25.26) | 52 (13.54) | 1 | 1 |
| | Married | 160 (41.67) | 61 (15.89) | 0.72 (0.42-1.25) | 0.90 (0.43-1.86) |
| | Divorced | 9 (2.34) | 5 (1.30) | 0.73 (0.19-2.84) | 1.90 (0.30-12.01) |
| Type of Profession | Nurse | 230 (59.90) | 89 (23.18) | 1 | 1 |
| | HO | 17 (4.43) | 9 (2.34) | 1.41 (0.51-3.88) | 0.67 (0.18-2.51) |
| | HIT | 7 (1.82) | 11 (2.86) | 4.64 (1.40-15.33) | 1.05 (0.23- 4.80) |
| | MD | 12 (3.13) | 9 (2.34) | 2.09 (0.74-5.91) | 1.11 (0.28- 4.44) |
| Work experience (in Years) | <=5 | 103 (26.82) | 49 (12.76) | 1 | 1 |
| | 6-10 | 116 (30.21) | 61 (15.89) | 0.91 (0.51-1.63) | 1.23 (0.53-2.87) |
| | >10 | 47 (12.24) | 8 (2.08) | 0.43 (0.17-1.09) | 0.74 (0.16-3.44) |
| Social media account ownership | No | 57 (14.84) | 2 (0.52) | 1 | 1 |
| | Yes | 209 (54.43) | 116 (30.21) | 29.92(6.32-141.51) | **13.96 (2.33-83.64)*** |
| Take ICT Training | No | 218 (56.77) | 51 (13.28) | 1 | 1 |
| | Yes | 48 (12.50) | 67 (17.45) | 4.31(2.42-7.68) | 1.81 (0.87-3.79) |
| Ever visited the medical field via mobile phone | No | 123 (32.03) | 13 (3.39) | 1 | 1 |
| | Yes | 143 (37.24) | 105 (27.34) | 7.03 (3.44-14.34) | **2.39 (1.03-5.51)*** |
| Digital Literacy | No | 47 (12.24) | 3 (0.78) | 1 | 1 |
| | Yes | 219 (57.03) | 115 (29.95) | 7.62 (2.18-26.66) | **6.13 (1.50- 25.01)*** |
| ESV-ICD-11 booklet guide awareness | No | 174 (45.31) | 58 (15.10) | 1 | 1 |
| | Yes | 92 (23.96) | 60 (15.63) | 4.18 (2.18-8.01) | **2.34 (1.06-5.18)*** |
| Facility type | Health Center | 173 (45.05) | 47 (12.24) | 1 | 1 |
| | Hospital | 93 (24.22) | 71 (14.49) | 3.58(0.73-17.65) | 1.39 (0.23-8.36) |
| Facility location | Rural | 93 (24.22) | 17 (4.43) | 1 | 1 |
| | Urban | 173 (45.05) | 101 (26.30) | 3.15 (0.84-11.85) | 2.88 (0.61-13.52) |
| Receive training/mentorship | No | 93 (24.22) | 17 (4.43) | 1 | 1 |
| | Yes | 173 (45.05) | 101 (26.30) | 5.78 (3.00-11.16) | **2.93 (1.25-6.87)*** |

Note: Bold values indicate significant. 1 = reference; * P-value <0.05, COR = Crude Odds Ratio, AOR = Adjusted Odds Ratio, CI = Confidence Interval; ICT = Information and Communication Technology; ESV-ICD-11 = Ethiopian Short Version of ICD-11; HO = Health officer; HIT = Health Information Technician; MD = Medical Doctor.

Finally, awareness of the Ethiopian Short Version ICD-11 (ESV-ICD-11) booklet was significantly associated with the awareness of the mobile-based app. The odds of healthcare providers being aware of the ESV-ICD-11 booklet were 3.35 times more likely to be aware of mobile-based disease classification app compared to those with no awareness of the ESV-ICD-11 booklet manual guide. As demonstrated by a study in U.S., this association may be explained by greater perceived usefulness and familiarity with the app's underlying classification system [37].

## Strength and limitation of the study

The strength of this study is the use of multilevel analysis to consider clustering and also conducted as multicenter settings including hospitals and health centers. Since the data was collected self-administered interview, they might be social desirability bias as a limitation of the study.

## Conclusion and recommendation

About one-third of healthcare providers were aware of the mobile-based disease classification (NHDD) app. Social media account ownership, Ever visited the medical field via mobile phone, digital literacy, familiarity with the Ethiopian Short-Version ICD-11 booklet, and receive training/mentorship were found to be significant factors for awareness of the app. These results underline the necessity of focused strategies to raise NHDD app awareness and use.

## Supporting information

**S1 Data. Dataset of healthcare providers' awareness of mobile-based app for disease classification in Central Gondar Zone healthcare facilities, 2023.**
(XLSX)

## Acknowledgments

We would like to extend our gratitude to the University of Gondar for their financial support to aid this research data collection. We also want to thank the Amhara Public Health Institute, the Central Gondar Zone health department, and the selected four district health office heads for their support in providing support letters by recognizing the benefits of this research. We thank study participants, data collectors, and supervisors. Finally, we offer our heartfelt appreciation to the University of Southeastern Norway for allowing me to participate in a student exchange program through the SEARCH project to cover travel and accommodation expenses.

## Author contributions

**Conceptualization:** Adamu Takele Jemere, Tesfahun Melese Yilma, Lemma Derseh Gezie, Binyam Tilahun.

**Data curation:** Adamu Takele Jemere.

**Formal analysis:** Adamu Takele Jemere, Tesfahun Melese Yilma, Lemma Derseh Gezie, Shegaw Anagaw Mengiste, Monika Knudsen Gullslett, Jens Johan Kaasbøll, Binyam Tilahun.

**Funding acquisition:** Binyam Tilahun.

**Methodology:** Adamu Takele Jemere, Tesfahun Melese Yilma, Lemma Derseh Gezie.

**Writing – original draft:** Adamu Takele Jemere, Tesfahun Melese Yilma, Binyam Tilahun.

**Writing – review & editing:** Adamu Takele Jemere, Tesfahun Melese Yilma, Lemma Derseh Gezie, Shegaw Anagaw Mengiste, Monika Knudsen Gullslett, Jens Johan Kaasbøll, Binyam Tilahun.

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
