## [Decision Letter · Decision Letter 0]

10 Dec 2025

PONE-D-25-34294Healthcare providers’ awareness of a mobile-based app for disease classification in northwest Ethiopia: A Multilevel Logistic Regression AnalysisPLOS One

Dear Dr. Jemere,

Thank you for submitting your manuscript to PLOS ONE. After careful consideration, we feel that it has merit but does not fully meet PLOS ONE’s publication criteria as it currently stands. Therefore, we invite you to submit a revised version of the manuscript that addresses the points raised during the review process.

If applicable, we recommend that you deposit your laboratory protocols in protocols.io to enhance the reproducibility of your results. Protocols.io assigns your protocol its own identifier (DOI) so that it can be cited independently in the future. For instructions see: https://journals.plos.org/plosone/s/submission-guidelines#loc-laboratory-protocols. Additionally, PLOS ONE offers an option for publishing peer-reviewed Lab Protocol articles, which describe protocols hosted on protocols.io. Read more information on sharing protocols at . Additionally, PLOS ONE offers an option for publishing peer-reviewed Lab Protocol articles, which describe protocols hosted on protocols.io. Read more information on sharing protocols at https://plos.org/protocols?utm_medium=editorial-email&utm_source=authorletters&utm_campaign=protocols..

We look forward to receiving your revised manuscript.

Kind regards,

Dereje Oljira Donacho, PhD

Academic Editor

PLOS One

Journal Requirements:

3. Please include captions for your Supporting Information files at the end of your manuscript, and update any in-text citations to match accordingly. Please see our Supporting Information guidelines for more information: http://journals.plos.org/plosone/s/supporting-information..

4. We are unable to open your Supporting Information file “NHDD Awareness.dta”. Please kindly revise as necessary and re-upload.

Reviewers' comments:

Reviewer's Responses to Questions

**Comments to the Author**

1. Is the manuscript technically sound, and do the data support the conclusions?

Reviewer #1: Yes

Reviewer #2: Yes

2. Has the statistical analysis been performed appropriately and rigorously? 

Reviewer #1: No

Reviewer #2: Yes

3. Have the authors made all data underlying the findings in their manuscript fully available?

Reviewer #1: No

Reviewer #2: Yes

4. Is the manuscript presented in an intelligible fashion and written in standard English?

Reviewer #1: Yes

Reviewer #2: No

5. Review Comments to the Author

Reviewer #1: First I would like to thanks to give me to put my expertise on this manuscript entitled” Healthcare providers’ awareness of a mobile-based app for disease classification in Northwest Ethiopia: A Multilevel Logistic Regression Analysis ”

General Question for the Couthers

As I have seen this manuscript is gives sound for the health sector. This is because currently Ethiopia uses ESV- ICD 11 and this is already launched and using practically so, what kinds of input dose this study add for those health care providers?

Specific Question

1.Do you think that currently is their M-based App functional in the health facility?

2.Form your regression you are using ICC how it could be? Do you think it is appropriate technique? Since your study participant were deferent levels

3.Form your result “Over half (55.73%) were aware with the ESV-ICD11's173 objective.” How this result could be? Because currently this system works for Medical doctors and HIT professionals, most of health care providers doesn’t have even information rather than awareness. Please clearly state this issue.

4.Have you seen the Category, Frequency and Percent from Table 2: Use ESV-ICD 11 booklet guide No = 37 and Yes = 115 and Had challenge when using NHDD mobile app No = 44 and Yes = 71 how this result is appeared make it clear?

5.From your result “Health care providers having social media accounts had substantially higher odds of awareness 207 (AOR = 13.96, 95% CI: 2.33–83.64, p < 0.01) than those without social media accounts.” could you interpreted the value of (AOR = 13.96, 95% CI: 2.33–83.64 the value of confidence interval and AOR is under question mark. I’m not clear such kinds of result.

6.From your Table 4: Multilevel Logistic regression analysis of Healthcare providers’ awareness of mobile-218 based app implementation for disease classification in Central Gondar Zone Healthcare: What is your outstanding reason to choice the reference? Make it scientific reason.

7.over half (55.73%) were aware with the ESV-ICD11's173 objective But From your conclusion one third of the health care provider were awarded based on your finding. How can you agree your report and your collusion please make it

Thank you for your interesting manuscript

Reviewer #2: I found this study very interesting, and I believe the outcomes have the potential to influence decision makers to facilitate and encourage wider use of mHealth applications among healthcare providers in Ethiopia. The findings highlight important gaps in awareness, and they can guide strategies to make such applications more accessible and integrated into clinical practice.

However, I would like to comment on two areas:

Study population: The study grouped together all healthcare providers, including nurses, doctors, technical assistants, and administrative staff. While I understand the intent to capture a broad perspective, it may not be entirely logical to include non-clinical staff, such as technical assistants and administrative personnel, since they are not the main target users of the NHDD mobile app. The analysis would be more focused and relevant if restricted to nurses and doctors, who are the primary groups that need—and should—use the application in daily practice.

Writing style: The overall clarity of the manuscript would benefit from further editing, particularly in the Results section. The presentation of findings sometimes lacks flow and precision, which may make it difficult for readers to follow the main outcomes. A clearer, more concise reporting of the statistical results would strengthen the impact of the paper.

6. PLOS authors have the option to publish the peer review history of their article (what does this mean?). If published, this will include your full peer review and any attached files.). If published, this will include your full peer review and any attached files.

.

Reviewer #1: No

Reviewer #2: **Yes:** Dr Reem Lafi AlmutairiDr Reem Lafi Almutairi

---

## [Author Response · Author response to Decision Letter 1]

16 Dec 2025

Here I attached the author's response to the reviewers' comments

---

## [Decision Letter · Decision Letter 1]

16 Apr 2026

Healthcare providers’ awareness of a mobile-based app for disease classification in northwest Ethiopia: A Multilevel Logistic Regression Analysis

PONE-D-25-34294R1

Dear Dr. Jemere,

We’re pleased to inform you that your manuscript has been judged scientifically suitable for publication and will be formally accepted for publication once it meets all outstanding technical requirements.

An invoice will be generated when your article is formally accepted. Please note, if your institution has a publishing partnership with PLOS and your article meets the relevant criteria, all or part of your publication costs will be covered. Please make sure your user information is up-to-date by logging into Editorial Manager at Editorial Manager® and clicking the ‘Update My Information' link at the top of the page. For questions related to billing, please contact  and clicking the ‘Update My Information' link at the top of the page. For questions related to billing, please contact billing support..

Kind regards,

Muhammad Farooq Umer, PhD Epidemiology and Health Statistics

Academic Editor

PLOS One

Additional Editor Comments (optional):

Reviewers' comments:

Reviewer's Responses to Questions

**Comments to the Author**

1. If the authors have adequately addressed your comments raised in a previous round of review and you feel that this manuscript is now acceptable for publication, you may indicate that here to bypass the “Comments to the Author” section, enter your conflict of interest statement in the “Confidential to Editor” section, and submit your "Accept" recommendation.

Reviewer #1: All comments have been addressed

Reviewer #2: (No Response)

2. Is the manuscript technically sound, and do the data support the conclusions?

Reviewer #1: Yes

Reviewer #2: Yes

3. Has the statistical analysis been performed appropriately and rigorously? 

Reviewer #1: Yes

Reviewer #2: Yes

4. Have the authors made all data underlying the findings in their manuscript fully available?

Reviewer #1: Yes

Reviewer #2: Yes

5. Is the manuscript presented in an intelligible fashion and written in standard English?

Reviewer #1: Yes

Reviewer #2: Yes

6. Review Comments to the Author

Reviewer #1: Thank you authors for addressing the comment. it is better to lookover the language.Thank you for addressing the comment. Better to address all the comments to increases the quality of the paper. Gramer, coherence and discussion part

This helps to increase the quality of the paper

Reviewer #2: I would like to commend the authors for addressing an important and relevant research topic. The focus on mobile health (mHealth) in Ethiopia, particularly in exploring factors beyond healthcare providers’ awareness, is timely and adds valuable insight to the field. The selection of the issue is well justified, and the manuscript demonstrates a thoughtful approach in identifying and examining the key contributing factors. Additionally, the hypotheses are clearly articulated, and the statistical analyses are appropriate and well-conducted. Overall, these aspects reflect a solid methodological foundation.

7. PLOS authors have the option to publish the peer review history of their article (what does this mean?). If published, this will include your full peer review and any attached files.). If published, this will include your full peer review and any attached files.

.

Reviewer #1: No

Reviewer #2: **Yes:** Reem Lafi AlmutairiReem Lafi Almutairi

---

## [Editor Report · Acceptance letter]

PONE-D-25-34294R1

PLOS One

Dear Dr. Jemere,

I'm pleased to inform you that your manuscript has been deemed suitable for publication in PLOS One. Congratulations! Your manuscript is now being handed over to our production team.

Kind regards,

on behalf of

Dr. Muhammad Farooq Umer

Academic Editor

PLOS One